# Developing a Swallow-State Monitoring System Using Nasal Airflow, Surface Electromyography, and Thyroid Cartilage Movement Detection

**DOI:** 10.3390/bioengineering11070721

**Published:** 2024-07-16

**Authors:** Wann-Yun Shieh, Mohammad Anwar Khan, Ya-Cheng Shieh

**Affiliations:** 1Department of Computer Science and Information Engineering, College of Engineering, Chang Gung University, No. 259, Wen-Hwa 1st Road, Kwei-Shan, Taoyuan 333, Taiwan; d1129003@cgu.edu.tw; 2Department of Physical Medicine and Rehabilitation, Chang Gung Memorial Hospital, 5 Fu-Hsing Street, Kwei-Shan, Taoyuan 333, Taiwan; 3Department of Computer Science, University of Illinois Urbana-Champaign, 201 North Goodwin Avenue, Urbana, IL 61801, USA; kshieh2@illinois.edu

**Keywords:** swallowing disorder, respiration and swallow coordination, noninvasive sensor, swallow state monitoring

## Abstract

The safe ingestion of food and water requires appropriate coordination between the respiratory and swallowing pathways. This coordination can be disrupted because of aging or various diseases, thereby resulting in swallowing disorders. No comparative research has been conducted on methods for effectively screening swallowing disorders in individuals and providing timely alerts to their caregivers. Therefore, the present study developed a monitoring and alert system for swallowing disorders by using three types of noninvasive sensors, namely those measuring nasal airflow, surface electromyography signals, and thyroid cartilage movement. Two groups of participants, one comprising healthy individuals (58 participants; mean age 49.4 years) and another consisting of individuals with a history of unilateral stroke (21 participants; mean age 54.4 years), were monitored when they swallowed five volumes of water. Through an analysis of the data from both groups, seven indicators of swallowing disorders were identified, and the proposed system characterized the individual’s swallowing state as having a green (safe), yellow (unsafe), or red (highly unsafe) status on the basis of these indicators. The results indicated that the symptoms of swallowing disorders are detectable. Healthcare professionals can then use these data to conduct assessments, perform screening, and provide nutrient intake suggestions.

## 1. Introduction

Swallowing is a complex process that relies on the coordination between nasal respiration, submental muscle response, and thyroid cartilage movement along the swallowing path. When this coordination is disrupted, swallowing disorders occur; in these disorders, food boluses or water may remain at various positions within the oral cavity, pharynx, or larynx. Numerous diseases, including neurological diseases, neuromuscular disorders, chronic indigestion disorders, gastroesophageal reflux disease, and head or neck cancers, can impair such coordination [1,2]. Aging is another factor that contributes to a decline in swallowing ability, with an estimated 10–30% of individuals aged over 65 years experiencing swallowing disorders [1].

Swallowing disorders directly affect nutrient intake and overall quality of life in older adults [3]. Many older adults frequently encounter difficulties in swallowing during meals and often require assistance from family members or caregivers. However, unless the individual exhibits explicit symptoms such as choking, coughing, or apnea, family members or caregivers can find it challenging to identify occurrences of swallowing disorders easily during meals. Furthermore, when appropriate treatment is not provided, older adults who experience a swallowing disorder are at risk of complications, including inhalation injuries, aspiration pneumonia, or even death [4].

Conventional methods for assessing swallowing disorders include videofluoroscopic swallowing studies (VFSSs) [5,6] and fiberoptic endoscopic evaluation of swallowing (FEES) [7]. VFSSs involve using X-ray videos to monitor the structural aspects of the swallowing process, such as the movements of the hyoid bone or thyroid cartilage. FEES involves inserting a fiberoptic endoscope through the nasal cavity to the pharynx to capture images of swallowing. Both approaches are invasive assessments and carry risks of radiation exposure or tissue injury for patients. Moreover, these assessments can only be conducted in specific laboratories or hospitals and may thus be inconvenient for individuals with limited mobility. The equipment required to evaluate swallowing disorders might also be unavailable at some healthcare facilities. Furthermore, the aforementioned approaches alone are insufficient for assessing the coordination between swallowing and respiration. Additional methods such as nasal airflow measurement, surface electromyography (sEMG), or swallowing sound analysis are often required to conduct a comprehensive evaluation.

Researchers have increasingly begun to use portable and noninvasive sensors, such as microphones [8], accelerometers [9], sEMG sensors [10], or bend sensors [11], in the assessment of various swallowing or ingestion behaviors. The use of such sensors offers advantages in terms of convenience, safety, and the absence of radiation exposure. Notably, our previous studies have indicated that the concurrent use of multiple sensors enables the comprehensive recording of the coordination between swallowing and respiration, which allows clinicians to track the entire swallowing process and evaluate oral and pharyngeal disorders [12,13,14].

Most studies on swallowing disorders have focused on swallowing evaluations, with limited studies addressing the monitoring and alert needs for elderly individuals with swallowing disorders. If an alert is provided prior to the onset of a swallowing disorder, the caregiver can take suitable actions, such as adjusting food hardness or reducing feeding speed, to prevent potential injuries like choking, aspiration, indigestion, etc. In addition, alerts can be provided remotely through telemonitoring when swallowing parameters can be measured using portable sensors. The demand for such alerts is increasing. Numerous clinical studies have highlighted that COVID-19 can result in swallowing disorders [15,16,17]. The results of these studies jointly suggest that the monitoring of swallowing disorder symptoms should be included in the treatment of patients with COVID-19 and in respiratory therapy to prevent further functional deterioration [17]. However, few studies have developed suitable methods for the remote monitoring of swallowing disorders.

The present study used multiple sensors to develop a monitoring and alert system for swallowing disorders. For convenience and effectiveness, we used three types of sensors, namely a pair of sEMG electrodes on the surface of submental muscles, an airflow cannula at the front of the nasal cavity, and a force-sensing resistor (FSR) on the surface of the thyroid cartilage, to obtain measurements on a range of parameters along the respiratory and swallowing paths. Temporal analysis was conducted on the signals detected by these sensors to assess respiratory–swallowing coordination in a non-invasive manner. Moreover, to identify suitable indicators of swallowing disorders, data were collected from two groups of participants: 58 healthy participants who had an average age of 49.4 years and 21 patients with unilateral stroke who had an average age of 54.4 years. Studies have indicated that patients with unilateral stroke tend to have not only swallowing disorders but also distinct swallowing patterns on several parameters [18,19]. Therefore, the swallowing parameters measured for these patients can be used as criteria for sending alerts related to incidents of swallowing disorders.

## 2. Materials and Methods

### 2.1. Proposed Alert Method for Swallowing Disorders

Figure 1 illustrates the proposed alert method for swallowing disorders. In this method, a swallow-state monitoring system with three types of noninvasive sensors is employed to measure swallowing parameters during swallowing when individuals consume food or water. Alerts are triggered according to a predefined set of indicators of respiratory–swallowing coordination. These indicators are identified on the basis of parameters recorded in the nasal, oral, and pharyngeal areas during the swallowing process.

The purpose of these alerts is twofold. First, they serve as prompts for caregivers, reminding them to consider adjustments to the patient’s eating or feeding strategies, which ensures the responsiveness and adaptability of patient care. Second, the alerts can be used as feedback in a swallowing training program under the guidance of qualified medical professionals. This feedback aids in refining and enhancing the efficacy of the swallowing training regimen.

The adopted sensor and the signal collection methods are detailed in the following sections.

### 2.2. Sensors

#### 2.2.1. sEMG Electrodes

sEMG was conducted by adhering a pair of bipolar electrodes (BIOPAC Systems, Goleta, CA, USA; Figure 2a) to both sides of the chin at the submentalis and a ground electrode on the neck near the chest (Figure 2b). sEMG signals were sampled at a frequency of 2 kHz and amplified by a factor of 1000 with a filter that restricted the bandwidth to 5–500 Hz. These signals were transmitted through a wireless electromyogram transmitter (BIOPAC Systems; Figure 2a) to a signal collector. The wireless transmitter could be fastened to a human arm.

#### 2.2.2. Nasal Airflow Cannula

A nasal airflow cannula (Salter Labs De Mexico, Chihuahua, Mexico; Figure 3a) was placed in front of the nasal cavity to detect respiration (Figure 3b), which comprises inhalation, exhalation, or apnea. The cannula was connected to a pressure transducer (Pro-Tech Services, Murrysville, PA, USA) that could convert airflow pressure into digital signals.

#### 2.2.3. Force-Sensing Resistor

An FSR is a type of piezoresistive sensor that is used to measure the force applied to a surface (Figure 4a). It typically consists of two layers: a conductive film and a semi-conductive polymer, separated by a thin spacer. The entire assembly is flexible, allowing it to be easily attached to various surfaces. In this study, an FSR was fixed over the larynx at the thyroid cartilage position. Our previous study [13] found that using only medical tape to attach the FSR to the larynx surface results in poor signal detection because medical tape cannot provide a sufficient reaction force on the FSR, making the sensor insensitive. Additionally, directly attaching the FSR to the larynx surface can obstruct natural swallowing motions, causing discomfort for the participant.

To address these issues, we used a throat-belt where the FSR is fixed on the center of the belt and the participant can wear it around the neck, as shown in Figure 4. The belt is elastic with soft cushioned material and secured with Velcro straps. With a maximum width of 5 cm, this design ensures minimal discomfort for the wearer and reduces the need for the belt to be tightened excessively. A small soft airbag is inserted between the belt and FSR, ensuring the FSR remains centered on the thyroid cartilage without movement during testing. When the belt is fastened around the neck, the airbag provides stable initial pressure on the FSR as the measurement baseline, ensuring sensor sensitivity. During swallowing, the thyroid cartilage moves up and then returns to its original position, altering the pressure on the FSR.

### 2.3. Signal Collector

The three types of sensors of the proposed system were connected to a signal collector to measure swallowing parameters (Figure 5). The signal collector had a length, width, and height of 102.9, 63.4, and 46.2 mm, respectively. Figure 5 illustrates the top and side views of the adopted signal collector. The core component of the collector was an Arduino UNO (Arduino, Ivrea, Italy) board that comprised a microcontroller unit, a 2-KB memory module, and data processing components. The collector reads sEMG signals using an RF module attached to the Arduino UNO device. The output port of the nasal airflow transducer has two wires that carry voltage signals. The first wire maintains a fixed voltage output, while the second wire’s voltage varies based on the pressure or force exerted during a breathing cycle. To monitor nasal airflow, the voltage difference between these two wires is computed. A Bluetooth module was also attached to the board for wireless communication. A mobile application was developed and installed on a remote tablet to display the data acquired by the collector for online analysis. The framework described in the aforementioned text was used to measure a set of parameters related to the coordination between respiration and swallowing in real-time.

### 2.4. Monitoring of and Alerts Related to Respiratory–Swallowing Coordination

#### 2.4.1. Parameter and Signal Analyses

Figure 6 displays the signals that were measured when a participant swallowed 5 mL of room-temperature water. The sEMG waveform indicated submentalis activity during swallowing (E1 to E3), with the largest amplitude being noted at E2. The nasal airflow waveform indicated a short period of apnea (A1 to A2), which is regarded as a protective phenomenon that enables safe swallowing without aspiration. The FSR waveform displayed a W-shaped response that represented two phases of thyroid cartilage movement. The first phase (F1 to F2) represented the movement of the thyroid cartilage upward and forward to block the trachea, and the second phase (F2 to F3) represented the thyroid cartilage movement that ensured that water smoothly passed through the pharynx to the esophagus. Finally, the thyroid cartilage returned to its original position. The total time from F1 to F3 is called the total excursion time (*TET*). Table 1 lists all parameters that describe a swallowing process along the respiratory and swallowing paths. In addition to the temporal parameters, two possible respiratory phases were observed before and after the respiration apnea period (A1 to A2), namely expiration (EXP) or inspiration (INP). Thus, pre-apnea/post-apnea respiratory patterns can come in one of four patterns: EXP/EXP, EXP/INP, INP/EXP, and INP/INP. Only EXP/EXP was regarded as a physiologically safe swallowing pattern [16] (the case in Figure 6 is EXP/INP).

#### 2.4.2. Swallowing Disorder Detection and Alert Indicators

Swallowing disorders can be detected by monitoring the parameters presented in Table 1. The alerts are classified into two risk level categories. Level 1 alerts are associated with two indicators: percentage of piecemeal deglutition (*PMD*) and long total excursion time (*L_TET*). When the volume of water or bolus exceeds a certain limit, individuals with insufficient swallowing strength break the water or bolus into smaller parts for swallowing (i.e., *PMD*). This phenomenon can be detected in FSR waveforms. In contrast to the typical W-shaped response for a single swallowing action, *PMD* is represented by a sequence of small pulses following the pulse corresponding to the first swallowing action. When the swallowing action is performed multiple times during a single instance of ingestion, the movement of the thyroid cartilage might persist for several seconds. If this occurs in an older adult with a swallowing disorder, a caregiver may need to stop the older adult from eating for a time. Failure to do so might result in the retention of excess water or bolus in the oral or pharyngeal area, which can potentially obstruct the respiratory passage and lead to choking. A similar situation occurs when the *TET* (i.e., *TET* in Table 1) for a single swallowing action exceeds a normal range. Research has indicated that *PMD* and *L_TET* disorders are highly common among individuals with swallowing difficulties [18,19]. In both cases, any eating or feeding should be paused temporarily to prevent unforeseen complications. The patient or caregiver should wait until the patient returns to a stable state before resuming the patient’s feeding process.

Level 2 alerts are associated with abnormal levels of the other parameters presented in Table 1. Four indicators are included in Level 2 alerts: (1) a long apnea duration (i.e., *L_SAD* in Table 2), (2) a large gap between the start times of sEMG signals and thyroid activity (i.e., *L_OL* in Table 2), (3) a long duration of physiological response during swallowing (i.e., *L_D_2DEF_* in Table 2), and (4) unsafe respiratory phases before and after respiration apnea (i.e., *U_Resp_phase* in Table 2). These indicators cover the coordination among nasal, oral, and pharyngeal processes and suggest a swallowing disorder.

The conditions for the triggering of Level 1 and Level 2 alerts are summarized in Table 2. This paper proposes a three-color alert model (Figure 7) with green, red, and yellow indicating safe, unsafe, and highly unsafe swallowing statuses, respectively. The alert state is in green when Level 1 or Level 2 indicators are absent. Figure 7 illustrates the transitions between the swallowing states. In this figure, green, yellow, and red represent a normal state, Level 2 alerts, and Level 1 alerts, respectively. Caregivers and patients can perform suitable actions during mealtime according to the color of the alert.

### 2.5. Participants: Healthy Controls and Patients with Unilateral Stroke

The participants of the test were selected from the rehabilitation department clinic at Chang Gung Memorial Hospital. A total of 58 healthy individuals (average age: 49.4 ± 15.6 years, male/female ratio: 28/30) and 21 patients with unilateral stroke (average age: 54.4 ± 10.5 years, male/female ratio: 18/3) participated in this study. The inclusion criteria for the healthy participants were as follows: (1) normal oral structure and normal oral function, (2) no history of swallowing impairment, and (3) no history of neurological or head and neck impairment that might affect swallowing function. In addition, patients with unilateral stroke were included if they had completed baseline and follow-up assessments in a rehabilitation department within 3 months after the diagnosis of stroke. Both groups were assessed using the Functional Oral Intake Scale (FOIS) [20] before the experiments of this study. All the healthy participants had FOIS levels exceeding 7, which indicated that they had the capacity for a complete oral diet with no restrictions. By contrast, the patients with unilateral stroke had FOIS levels ranging from 4 to 7, which suggested that they could manage a complete oral diet but required special dietary preparation. Table 3 lists the characteristics of the participants.

Each participant was asked to swallow 1, 3, 5, 10, and 20 mL of room-temperature water. The testing for each volume was repeated thrice. The participants rested for 2 min after swallowing a given water sample prior to doing so for the next. To ensure participant safety, testing was initiated with the smallest water volume, and the volume was progressively increased. Before testing commenced, each participant received a detailed explanation regarding the present study’s objectives and the testing procedure. Moreover, written informed consent was obtained from all participants prior to the tests. This study adhered to the ethical guidelines outlined in the Declaration of Helsinki and received approval from the Institutional Review Board of Chang Gung Memorial Hospital, Taoyuan, Taiwan (protocol code: 201800480B0; date of approval: 1 August 2018).

## 3. Results

The effectiveness of each alert indicator was evaluated by comparing the measurement results obtained for the healthy participants and patients. Statistical analyses were conducted using the SPSS 12.0 software program (SPSS, Chicago, IL, USA). The data obtained from the three swallowing trials for each volume of water were averaged. Paired samples *t*-tests were performed to examine the significance of the differences between the two groups.

### 3.1. Verification of the Signal Processing

Table 4 presents the error rates observed for the data collected using the signal collector compared with the data collected using a BIOPAC MP150 biosignal detector (BIOPAC Systems) with the same sensors as the signal collector. Higher error rates were observed in the results obtained by the nasal airflow and FSR sensors than in those obtained by the sEMG electrodes. This disparity was likely caused by the greater complexity of patterns in nasal airflow and FSR signals than in sEMG signals. Other possible factors responsible for the aforementioned disparity were the limited computing power of the signal collector and its lower sampling rate (1 kHz) compared with that of the biosignal detector (typically > 10 kHz). Nevertheless, all error rates were below 5%, which indicated satisfactory accuracy.

### 3.2. Level 1 Alerts

#### 3.2.1. Piecemeal Deglutition

Table 5 presents the *PMD* values for the healthy and patient groups. The *PMD* value exhibited a proportional increase with the water volume for both groups. As the water volume exceeded 5 mL, the *PMD* values for the healthy and patient groups surpassed 10% and 20%, respectively. A water volume between 5 and 10 mL is safe to swallow [12]. The healthy group had appropriate *PMD* values for a water volume of up to 10 mL. However, the patient group exhibited a noticeable increase in their *PMD* values as the water volume was increased to 3 mL (i.e., 17.4%) or more. Thus, patients with unilateral stroke faced difficulties in swallowing. The results highlight the importance of administering such patients small volumes of water (<5 mL) to ensure safe swallowing.

The *PMD* value for the patient group consistently exceeded that for the healthy group across all water volumes. A paired samples *t*-test demonstrated that the *PMD* differences between the two groups were significant, with all *p* values being < 0.05. These results underscore that patients with unilateral stroke have significantly higher *PMD* values than healthy individuals; thus, *PMD* is a crucial indicator of swallowing disorders.

#### 3.2.2. Total Excursion Time

Figure 8 illustrates the *TET* results for both groups when cases of *PMD* were ignored. Instances of piecemeal deglutition were ignored because in most of these instances, the participants required more than twice the normal time to complete the swallowing process.

The *TET* value increased with the water volume for both groups. The patient group generally exhibited larger *TET* values than did the healthy group for all water volumes. However, for water volumes of ≥5 mL, the disparity between the two groups increased considerably. The patient group required more than 1.2 s to successfully swallow 10 and 20 mL of water. This result indicates that patients with unilateral stroke should reduce their swallowing speed and pay attention to the amount of water intake according to their individual condition.

Table 6 indicates the percentages of participants with TET (*L_TET*) values larger than a predefined threshold (*T*). The threshold value (*T*), which was set as the mean *TET* value of the healthy group, served as a benchmark for evaluating swallowing ability. Two key observations indicated that *L_TET* could be used to effectively distinguish the swallowing abilities of the two groups. First, the patient group consistently exhibited significantly higher *L_TET* values across all volumes than did the healthy group (*p* < 0.05). Second, in the swallowing tests conducted with 1 and 20 mL of water, the *L_TET* values for the patient group exceeded 90%. These volumes were also challenging for most of the healthy participants to swallow, which reinforced the finding that the patients had to exert greater effort to complete the 1- and 20-mL swallowing tests than the other two tests.

### 3.3. Level 2 Alerts

#### 3.3.1. Duration of Respiration Apnea

Figure 9 displays the durations of respiration apnea (*SAD*) for the two groups. In contrast to the disparities in the Level 1 indicators, the *SAD* disparities between the two groups were generally modest, with noticeable differences primarily evident in the 3- and 5-mL tests. The results presented in Table 7 indicate that 71% of the patient group had *SAD* values higher than the mean *SAD* of the healthy group in the 3- and 5-mL tests. This result suggests that a large proportion of the patient group required an extended respiratory pause to ensure the safe swallowing of normal water volumes. This phenomenon, which appears to be a compensatory protective response caused by functional deterioration, was more frequently observed in the patient group than in the healthy group. For the 1-, 10-, and 20-mL tests, the *SAD* values of the patient group were nonsignificantly higher than the mean *SAD* of the healthy group.

#### 3.3.2. Onset Latency between sEMG Signals and Thyroid Cartilage Excursion

Figure 10 displays the mean *OL* values for the healthy and patient groups. Effective swallowing necessitates close coordination between the submental muscles and the thyroid cartilage; therefore, the onset latency (*OL*) between these structures is a crucial indicator of swallowing coordination [14]. The results displayed in Figure 10 reveal that for all test volumes, the patients exhibited larger *OL* values than the healthy participants. In particular, notable differences were observed between the two groups for volumes of 3–10 mL (Table 8), and the differences began to diminish as the volume was increased beyond 10 mL. Nevertheless, 80.9% of the patient group exhibited higher *OL* values than the mean *OL* value of the healthy group in the 20-mL test. This result indicated that a considerable number of patients always exhibited uncoordinated swallowing, particularly when swallowing large volumes of water.

#### 3.3.3. Long Duration of the Second Deflection in the W-Shaped Response of the FSR

Figure 11 displays the mean durations of the second deflection in the W-shaped response of the FSR (*D_2DEF_*) for the healthy and patient groups. The parameter *D_2DEF_* represents the duration for which the thyroid cartilage squeezes water into the esophagus and then returns to its initial position. A high *D_2DEF_* value indicates a high risk of water retention in the pharynx. The results depicted in Figure 11 reveal that for all water volumes, the mean *D_2DEF_* value for the patient group exceeded that for the healthy group. Table 9 presents the percentages of participants in each group with *D_2DEF_* values higher than a threshold (i.e., the mean *D_2DEF_* value for the healthy group). The results indicated that for all water volumes, the aforementioned percentage for the patient group was significantly higher than that for the healthy group. This observation can be attributed to most patients exerting force over an extended period when they swallowed water at each volume, which caused fatigue in the thyroid cartilage. This fatigue then resulted in a compensatory mechanism that led to a high *D_2DEF_* value.

#### 3.3.4. Long Durations of sEMG Signals for Submentalis Activity

Figure 12 displays the mean durations of sEMG signals for submentalis activity (*D_sEMG_*) for the healthy and patient groups. A high *D_sEMG_* value indicates an extended total duration spent pushing water from the oral stage to the pharyngeal stage. For all volumes of water, the patient group exhibited a marginally higher *D_sEMG_* value than the healthy group. This result indicates that on average, patients with unilateral stroke have a weaker ability to contract submentalis than do healthy individuals; thus, these patients require more time to swallow water safely.

Table 10 presents the percentages of participants with *D_sEMG_* values that exceeded the mean *D_sEMG_* value of the healthy group. A total of 61.9–76.1% of the patient group exhibited *D_sEMG_* values higher than the mean *D_sEMG_* value of the healthy group; the corresponding percentage for the healthy group was below 50%. These results are consistent with those obtained for other Level 2 parameters (i.e., *L_SAD*, *L_OL*, and *L_D_2DEF_*) and indicated a uniform presentation of swallowing disorders among the patients.

#### 3.3.5. Unsafe Respiratory Phases before and after Respiration Apnea

Table 11 presents the proportions of healthy participants and patients with safe respiratory phases before and after swallowing each volume of water (EXP/EXP). Notably, more than 59% of the healthy participants exhibited EXP/EXP phases, with the corresponding value for the patients being approximately 50%. The results of a *t*-test (last column of Table 11) indicated that the two groups exhibited significant differences in the aforementioned ratio in the 1-, 3-, and 5-mL tests. The EXP/EXP respiratory phase is considered a protective respiratory pattern before and after swallowing. Compared with the healthy group, a higher proportion of the patient group tended to remain in nonprotective respiratory phases during swallowing. Thus, *U_Resp_phase* is a suitable indicator for identifying swallowing disorders.

## 4. Discussion

The proposed system has two Level 1 alerts, one based on *PMD* and the other based on *L_TET*. The experimental results of this study revealed significant differences in PMD and *L_TET* between healthy individuals (the healthy group) and individuals with unilateral stroke (the patient group). For water volumes of 1–5 mL, the *PMD* values for the healthy group ranged from 0.5% to 3.4%. Moreover, for water volumes of 10 and 20 mL, the *PMD* value for the healthy group was 11.4% and 23.8%, respectively. Nevertheless, for all water volumes, the *PMD* value for the healthy group was substantially lower than that for the patient group (11.2–31.4% for water volumes of 1–20 mL). The *L_TET* values of the two groups also differed significantly. The healthy group had *L_TET* values of 37.9–44.8%, and the patient group had *L_TET* values of 71.4–90.5%. The aforementioned results validate the effectiveness of *PMD* and *L_TET* as strong indicators of swallowing disorders. The triggering of the *PMD* and *L_TET* alerts during a swallow suggests that the swallowing pathway is severely obstructed, which indicates that the eating or feeding strategy should be modified to prevent unexpected injuries.

The proposed system has four Level 2 alerts that are based on *L_SAD*, *L_OL*, *L_ D_2DEF_*, and *U_Resp_phase*. In contrast to the Level 1 alerts, the Level 2 alerts exhibited the most considerable difference between the two groups for water volumes of 3–10 mL. Although significant differences were not observed for all water volumes, the high values of the Level 2 alerts might have still had negative effects on the patients. Therefore, a flexible and adaptable monitoring and alert mechanism must be established to cater to the individual needs of each patient. The mechanism proposed in this paper offers such flexibility by allowing for customizing the transition conditions between the states in Figure 7.

In this study, swallowing signals were captured using three types of sensors (a nasal cannula, sEMG electrodes, and an FSR). All materials selected in this study were soft, flexible, and designed for easy attachment, thereby ensuring that they did not impede the participants’ natural swallowing function. Furthermore, signal collection and processing were conducted using a compact handheld device. The ability to detect and display patterns of swallowing disorder signals was integrated into an app, thereby enhancing the proposed system’s portability and user-friendliness. This design makes the proposed system accessible to a wide range of participants, particularly those with limited mobility. Table 12 shows the comparison of the commercial devices or existing sensors with the sensors used in this paper.

The proposed system not only transmits essential data for the real-time monitoring and alerting of swallowing disorders but also can document a patient’s respiratory and swallowing coordination over time. This feature aids in the long-term evaluation of eating behaviors and swallowing safety. Caregivers can monitor the occurrences of Level 1 and Level 2 alerts, which provide valuable insights into the patient’s condition. Moreover, physicians can leverage the data collected by the proposed system to identify early indicators of swallowing difficulties and conduct comprehensive functional assessments.

The current study has some limitations. First, only three types of sensors were used for data collection in this study; other relevant research has used a wider array of sensors, such as microphones and accelerometers, to capture various dimensions of swallowing responses. The proposed signal collector was closely integrated with a monitoring and alert application, which prevented the incorporation of additional types of sensors into the proposed system.

Second, the sensors used in this study were not validated against VFSSs or FEES, which are considered to be more precise methods of assessment. Thus, potential uncertainties exist regarding the accuracy and reliability of the collected sensor data. Despite this limitation, the proposed monitoring system is a useful tool for the preliminary screening of individuals at risk of dysphagia at the bedside. Individuals identified as having a high risk of dysphagia can be subsequently directed for more comprehensive evaluations to confirm dysphagia and establish an appropriate treatment plan.

Finally, this study involved 58 healthy individuals and 21 patients with a history of unilateral stroke; thus, the sample of this study was imbalanced. This limitation was primarily caused by difficulties in recruiting a sufficient number of eligible patients. Furthermore, the scope of the tests was limited to swallowing water, with the swallowing of other types of foods (such as soft foods) or different bolus materials not being examined. The proposed monitoring system’s capability to assess swallowing function should be thoroughly evaluated for other types of foods and different bolus materials. In addition, the used time length of the proposed system and the stability of the sensors over a certain period were not addressed in this study and should be considered important topics for future research.

## 5. Conclusions

This study developed a noninvasive system for monitoring and providing alerts regarding respiratory–swallowing coordination. This system employs three types of sensors to monitor swallowing responses: a nasal airflow cannula (to measure nasal airflow), sEMG electrodes (to monitor submental muscle response), and an FSR (to monitor thyroid cartilage movement). The system’s main features include offering instant feedback on swallowing efficiency, supplying critical information regarding respiratory–swallowing coordination, and sending alerts to caregivers. It can also log meal intake, collect and transmit data to cloud-based databases, and facilitate health professionals in performing evaluations, screenings, and care consultations for individuals requiring assistance.

This study provides a series of effective indicators for detecting swallowing disorders. Data on these indicators can be used as a reference for enhancing the eating or feeding habits of the elderly or those with disabilities. Moreover, the collected data can be consolidated and archived in big-data systems, thereby paving the way for the application of these data in machine-learning technologies. As algorithms continue to evolve, such aggregated data can be used to create a semiautomatic system for analyzing swallowing functions, thereby simplifying the processes of early screening and safety monitoring.

## Figures and Tables

**Figure 1 bioengineering-11-00721-f001:**
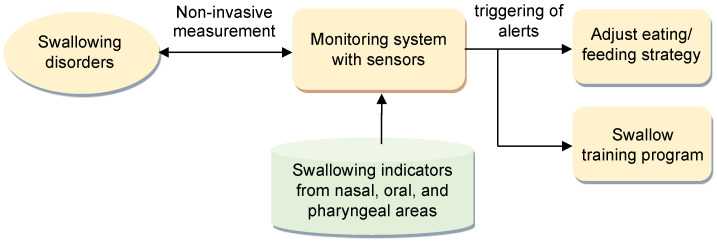
Proposed alert method for swallowing disorders.

**Figure 2 bioengineering-11-00721-f002:**
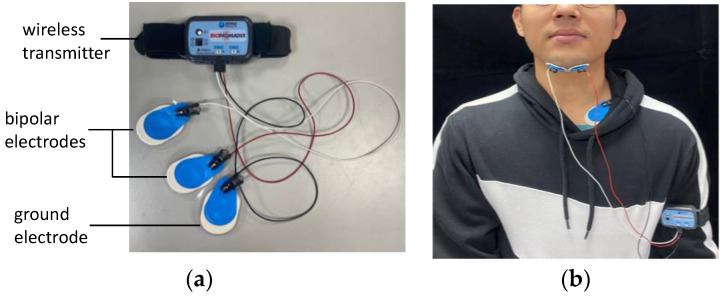
Surface electromyography (sEMG) measurement: (**a**) bipolar electrodes and (**b**) attachment of the sEMG electrodes to the human body.

**Figure 3 bioengineering-11-00721-f003:**
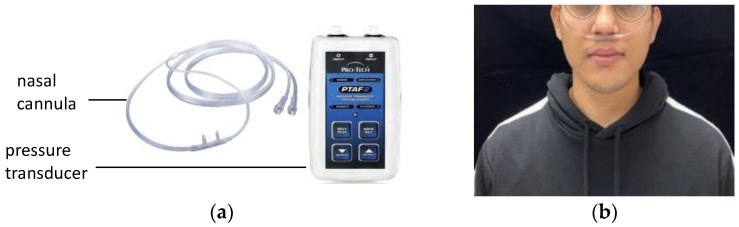
Nasal airflow measurement: (**a**) cannula and pressure transducer and (**b**) placement of the nasal cannula.

**Figure 4 bioengineering-11-00721-f004:**
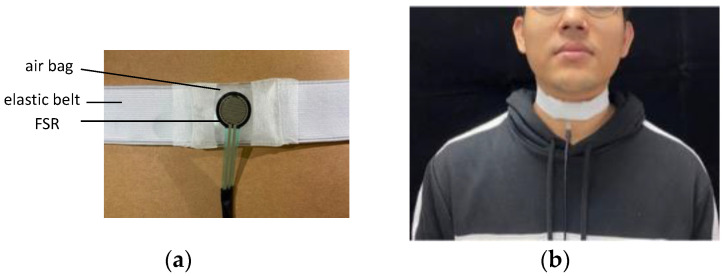
Measurement with the FSR: (**a**) FSR with an elastic belt and (**b**) placement of the FSR belt.

**Figure 5 bioengineering-11-00721-f005:**
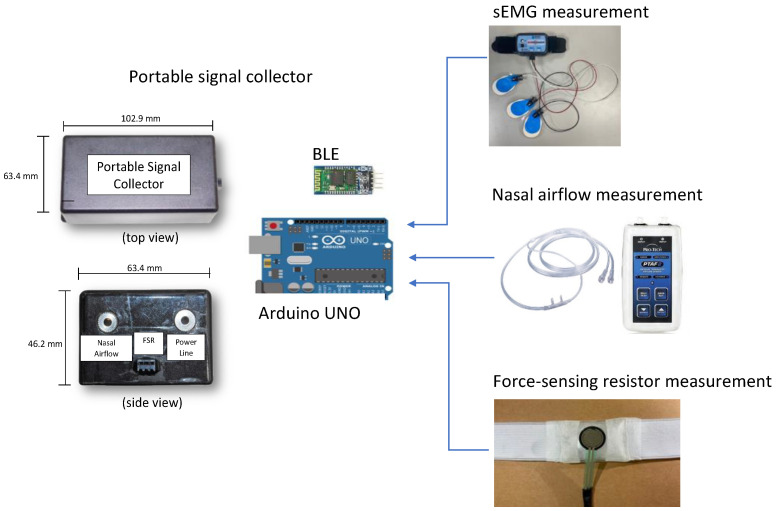
Design of the adopted signal collector, including the top view and side view.

**Figure 6 bioengineering-11-00721-f006:**
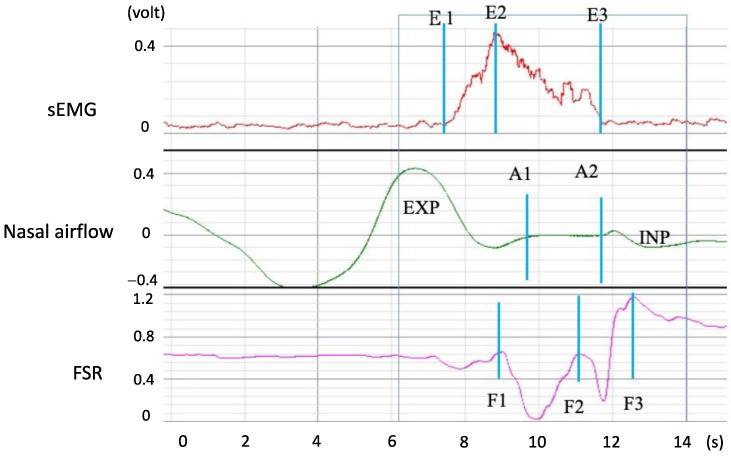
Swallowing and respiration signals acquired by the three types of adopted sensors. In this figure, E1–E3, A1 to A2, F1 to F2, and F2 to F3 represent durations of the (1) submental muscle response, (2) respiration apnea, (3) upward and forward movement of the thyroid cartilage to block the trachea, and (4) movement of the thyroid cartilage back to its original position, respectively.

**Figure 7 bioengineering-11-00721-f007:**
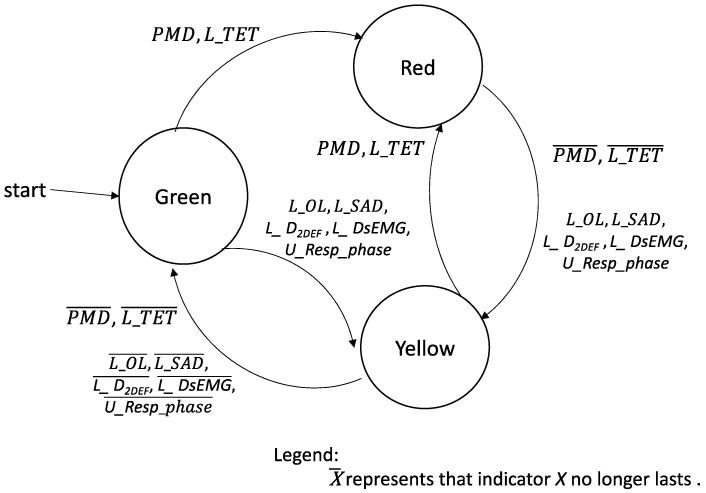
Three-color alert model for the swallowing state. Green, yellow, and red correspond to a normal state, Level 2 alerts, and Level 1 alerts, respectively.

**Figure 8 bioengineering-11-00721-f008:**
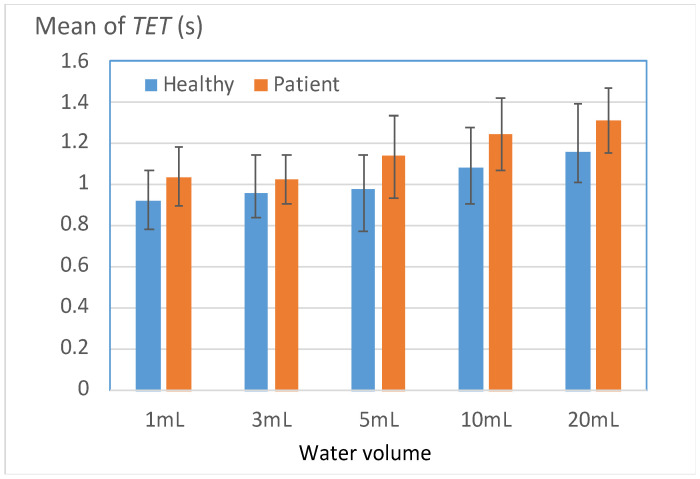
Mean total excursion time (*TET*) values for the healthy and patient groups.

**Figure 9 bioengineering-11-00721-f009:**
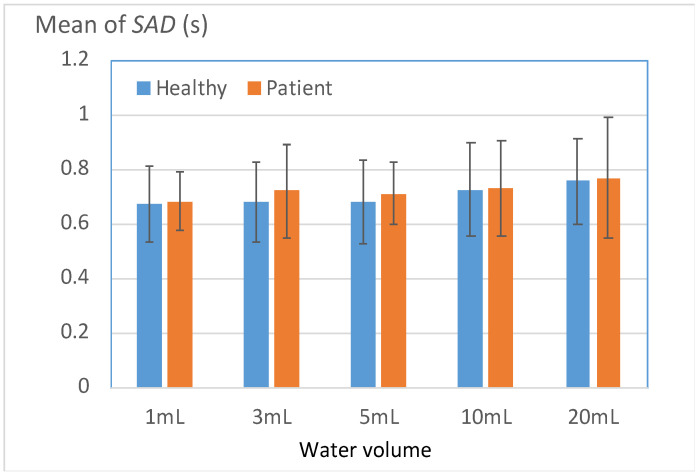
Mean durations of respiration apnea (*SAD*) for the patient and healthy groups.

**Figure 10 bioengineering-11-00721-f010:**
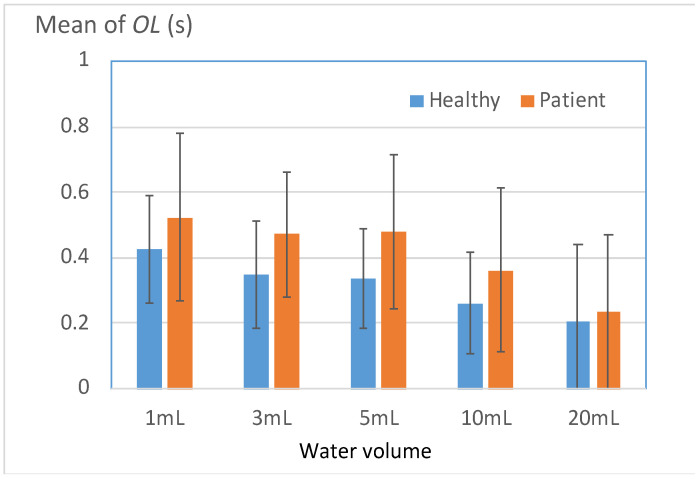
Mean onset latency between sEMG signals and thyroid cartilage excursion (*OL*) for the two groups.

**Figure 11 bioengineering-11-00721-f011:**
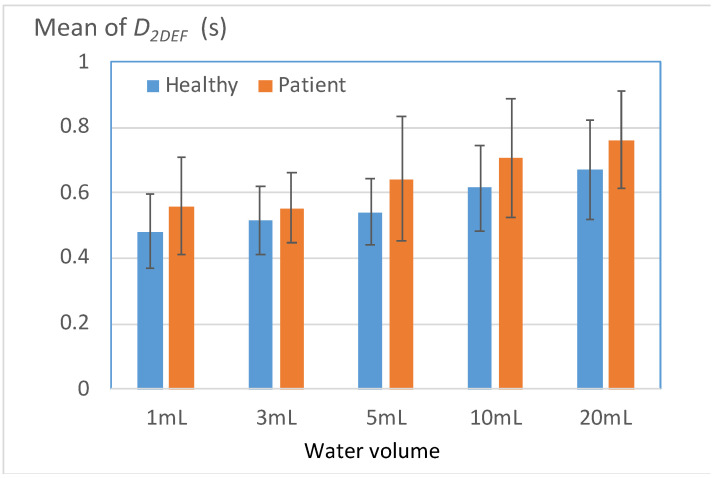
Mean durations of the second deflection in the W-shaped response of the FSR (*D_2DEF_*) for the two groups.

**Figure 12 bioengineering-11-00721-f012:**
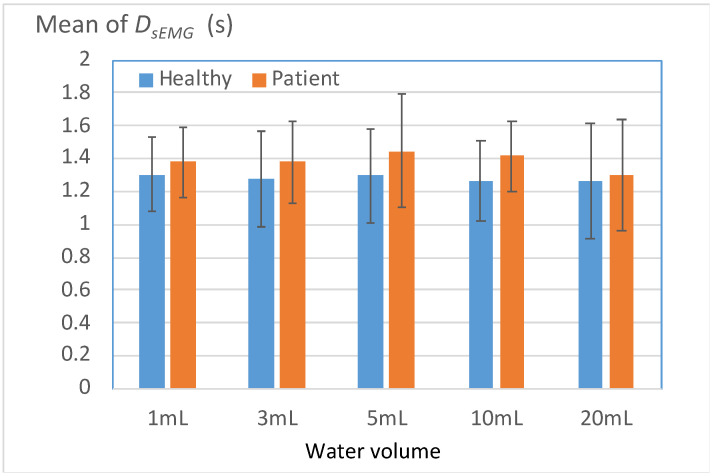
Mean durations of sEMG signals for submentalis activity (*D_sEMG_*) for the two groups.

**Table 1 bioengineering-11-00721-t001:** Parameters obtained from the respiration and swallowing signals.

Parameters	Definition	Calculation
*D_sEMG_*	Duration of submentalis surface EMG	E1 to E3
*SAD*	Duration of respiration apnea by nasal airflow	A1 to A2
*OL*	Onset latency between sEMG and thyroid cartilage movement	F1 to E1
*TET*	Total excursion time	F1 to F3
*D_2DEF_*	Duration of the 2nd deflection in the W-shaped response of FSR	F2 to F3
*Resp_phase*	Respiratory phases before/after SAD (i.e., EXP/EXP, EXP/INP, INP/EXP, or INP/INP)	State before/after swallow apnea

**Table 2 bioengineering-11-00721-t002:** Definition of risk levels. The last column indicates the criteria for the triggering of alerts.

Risk Level	Indicators	Definition	Triggering
Level 1 (Red alert)	*PMD*	Piece-meal deglutition	*TET* > *T ** and Non-W-shaped response
	*L_TET*	Long total excursion time of thyroid cartilage	*TET* > *T **
Level 2 (Yellow alert)	*L_SAD*	Long apnea duration	*SAD* > *T **
	*L_OL*	Long onset latency between sEMG and thyroid cartilage excursion	*OL* > (1/2) *T **
	*L_ D_2DEF_*	Long duration of second deflection in the W-shaped response of the thyroid cartilage	*D_2DEF_ *> (1/2)*T **
	*L_ D_sEMG_*	Long duration of submentalis surface EMG activity	*D_sEMG_* > *T **
	*U_Resp_phase*	Unsafe respiratory phases before/after SAD	None-EXP/EXP phases before A1 and after A2

(* Note. *T*: a predefined time threshold).

**Table 3 bioengineering-11-00721-t003:** Characteristics of the participants.

Group	Age Range (Years)	Age Mean ± SD (Years)	Male	Female	FOIS
Healthy	26–77	49.4 ± 15.6	28	30	Level 7
Ever unilateral stroke	39–78	54.4 ± 10.5	18	3	4≤ Level <7

**Table 4 bioengineering-11-00721-t004:** Error rates for the data collected with the signal collector compared with the data collected using the BIOPAC MP150 biosignal detector.

	1 mL	3 mL	5 mL	10 mL	20 mL
sEMG	0.79%	0.72%	0.74%	0.74%	0.74%
Nasal	2.56%	4.24%	1.05%	4.33%	3.18%
FSR	3.25%	2.35%	1.1%	2.98%	1.35%

**Table 5 bioengineering-11-00721-t005:** Percentage of piecemeal deglutition in the healthy and patient groups.

Volume	Group	Percentage	*p*-Value
1 mL	Healthy	0.5%	0.001 *
	Patient	11.2%	
3 mL	Healthy	2.2%	0.001 *
	Patient	17.4%	
5 mL	Healthy	3.4%	0.003 *
	Patient	14.2%	
10 mL	Healthy	11.4%	0.012 *
	Patient	26.9%	
20 mL	Healthy	23.8%	0.023 *
	Patient	31.4%	

(* *p* < 0.05).

**Table 6 bioengineering-11-00721-t006:** Percentages of participants with *TET* values larger than a predefined threshold *T*.

Volume	Mean of the Healthy Group, *T* (s)	Group	Percentages > *T*	*p*-Value (* *p* < 0.05)
1 mL	0.921	Healthy	41.3%	0.005 *
		Patient	90.5%	
3 mL	0.962	Healthy	44.8%	0.018 *
		Patient	71.4%	
5 mL	0.978	Healthy	43.1%	0.001 *
		Patient	71.4%	
10 mL	1.080	Healthy	41.3%	0.003 *
		Patient	80.9%	
20 mL	1.160	Healthy	37.9%	0.032 *
		Patient	90.5%	

**Table 7 bioengineering-11-00721-t007:** Percentages of participants with *SAD* values higher than a predefined threshold *T.*

Volume	Mean of Healthy Group, *T* (s)	Group	Percentages > *T*	*p*-Value (* *p* < 0.05)
1 mL	0.674	Healthy	48.3%	0.371
		Patient	52.4%	
3 mL	0.681	Healthy	50.1%	0.002 *
		Patient	71.4%	
5 mL	0.680	Healthy	43.1%	0.001 *
		Patient	71.4%	
10 mL	0.726	Healthy	43.2%	0.392
		Patient	42.9%	
20 mL	0.756	Healthy	41.3%	0.421
		Patient	52.4%	

**Table 8 bioengineering-11-00721-t008:** Percentages of participants with *OL* values higher than a predefined threshold *T*.

Volume	Mean of Healthy Group, *T* (s)	Group	Percentages > *T*	*p*-Value (* *p* < 0.05)
1 mL	0.424	Healthy	46.5%	0.053
		Patient	66.7%	
3 mL	0.348	Healthy	44.8%	0.008 *
		Patient	71.4%	
5 mL	0.336	Healthy	50.0%	0.004 *
		Patient	71.4%	
10 mL	0.261	Healthy	46.5%	0.046 *
		Patient	66.7%	
20 mL	0.205	Healthy	34.4%	0.693
		Patient	80.9%	

**Table 9 bioengineering-11-00721-t009:** Percentages of participants with *D_2DEF_* values higher than a predefined threshold *T.*

Volume	Mean of Healthy Group, *T* (s)	Group	Percentages > *T*	*p*-Value (* *p* < 0.05)
1 mL	0.482	Healthy	50.0%	0.019 *
		Patient	66.7%	
3 mL	0.518	Healthy	44.8%	0.213
		Patient	61.9%	
5 mL	0.541	Healthy	43.1%	0.005 *
		Patient	71.4%	
10 mL	0.615	Healthy	44.8%	0.028 *
		Patient	75.0%	
20 mL	0.671	Healthy	34.4%	0.049 *
		Patient	80.9%	

**Table 10 bioengineering-11-00721-t010:** Percentages of participants with *D_sEMG_* values higher than a predefined threshold *T*.

Volume	Mean of Healthy Group, *T* (s)	Group	Percentages > *T*	*p*-Value (* *p* < 0.05)
1 mL	1.302	Healthy	48.2%	0.208
		Patient	66.7%	
3 mL	1.275	Healthy	41.3%	0.186
		Patient	61.9%	
5 mL	1.294	Healthy	41.3%	0.065
		Patient	71.4%	
10 mL	1.264	Healthy	43.1%	0.027 *
		Patient	76.1%	
20 mL	1.258	Healthy	32.7%	0.695
		Patient	76.1%	

**Table 11 bioengineering-11-00721-t011:** Proportions of healthy participants and patients with safe respiratory phases before and after swallowing each volume of water [expiration (EXP)/EXP)].

Water Volume	Ratio of *Resp_Phase*: EXP/EXP		*p*-Value
	Healthy	Patient	
1 mL	69.8%	45.5%	0.001 *
3 mL	64.7%	54.5%	0.031 *
5 mL	64.3%	50.9%	0.027 *
10 mL	59.0%	53.2%	0.073
20 mL	60.8%	47.2%	0.060

(* *p* < 0.05).

**Table 12 bioengineering-11-00721-t012:** Comparison of the commercial or existing devices (sensors) with the sensors used in this paper.

	Measured Functions	Invasion	Sensing	Medical Expenses
FOIS [20]	Chewing/ Swallowing	Non-invasive	Use the questionary to assess swallowing function	Low
VFSS [5,6,11,21]	Swallowing	Invasive	Swallow barium-coated food and liquid, which are visible on X-ray	High
FEES [7]	Swallowing	Invasive	Use an endoscope inserted through the nose to view the pharynx and larynx	High
Sound Sensor [8,22,23]	Chewing/ Swallowing	Non-invasive	Use microphone to detect laryngeal sounds	Low
Strain Sensor [8,11,21,24]	Chewing	Non-invasive	Use the sensor to measure the amount of deformation in an object (length or bending angle)	Low
Accelerometry [9,22,23]	Chewing/ Swallowing	Non-invasive	Use accelerometer to detect cervical vibration	Low
sEMG [10,12,13,14,18,19,25] *	Chewing/ Swallowing	Non-invasive	Use electrodes to detect electrical signals from skeletal muscles	Low
FSR [13,14,18,19] *	Swallowing	Non-invasive	Use conductive layers to measure the applied force or pressure	Low
Nasal airflow [9,12,13,14,18,19] *	Respiration/ swallowing	Non-invasive	Use airflow cannula at the front of nasal cavity to measure airflow	Low

(* sensors used in this paper).

## Data Availability

https://reurl.cc/4rZWGL, accessed on 3 June 2024.

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
