# Peer review of "Developing a Swallow-State Monitoring System Using Nasal Airflow, Surface Electromyography, and Thyroid Cartilage Movement Detection"

_bioengineering, 2024, doi:10.3390/bioengineering11070721_

Round 1
Reviewer 1 Report (Previous Reviewer 2)
Comments and Suggestions for Authors
Thank you to all the authors for revising the manuscript according to the reviewer's concerns. Regarding my previous comment, I still have a concern with Figure 2. From Figure 2a, it appears that the white-colored electrode is the ground, while the red and black ones are bipolar electrodes. However, the white and red electrodes are affixed to the chin, and the text states, 'sEMG was conducted by adhering a pair of bipolar electrodes (BIOPAC Systems, Goleta, CA, USA; Figure 2a) to both sides of the chin at the submentalis...'. Please clarify which electrodes are placed where and ensure the text accurately reflects this.
Comments on the Quality of English LanguageThank you to all the authors for revising the manuscript according to the reviewer's concerns. Regarding my previous comment, I still have a concern with Figure 2. From Figure 2a, it appears that the white-colored electrode is the ground, while the red and black ones are bipolar electrodes. However, the white and red electrodes are affixed to the chin, and the text states, 'sEMG was conducted by adhering a pair of bipolar electrodes (BIOPAC Systems, Goleta, CA, USA; Figure 2a) to both sides of the chin at the submentalis...'. Please clarify which electrodes are placed where and ensure the text accurately reflects this.
Author Response
Comment 1:
Thank you to all the authors for revising the manuscript according to the reviewer's concerns. Regarding my previous comment, I still have a concern with Figure 2. From Figure 2a, it appears that the white-colored electrode is the ground, while the red and black ones are bipolar electrodes. However, the white and red electrodes are affixed to the chin, and the text states, 'sEMG was conducted by adhering a pair of bipolar electrodes (BIOPAC Systems, Goleta, CA, USA; Figure 2a) to both sides of the chin at the submentalis...'. Please clarify which electrodes are placed where and ensure the text accurately reflects this.
Response 1:
Thank you very much for pointing out the electrode problem in Figure 2. In Figure 2a, the original labels of the electrodes are wrong. The red and white-colored electrodes are the bipolar electrodes, and the black electrode is the ground (i.e., Figure 2b is correct). We have revised Figure 2a. Please see Figure 2 on Page 3.
Reviewer 2 Report (New Reviewer)
Comments and Suggestions for Authors
Figure 8 and table 7 with non-significant results are unnecessary.
Small number of patients, increase to at least 30.
Replace references 5 and 6 with newer ones.
Author Response
Thank you very much for the further comments and suggestions.
Comment 1: Figure 8 and table 7 with non-significant results are unnecessary.
Response 1:
To provide a complete comparison and discussion, it is necessary to include the non-significant results in Figure 8 and Table 7. Please check Figure 8 (P.11) and Table 7 (P.12).Comment 2: Small number of patients, increase to at least 30.
Response 2:
The number of patients in this study is indeed a limitation due to the difficulty of recruiting participants. We have addressed this issue in Section 4, "Discussion." Please refer to Lines 484-486 on page 18.
Comment 3: Replace references 5 and 6 with newer ones.
Response 3:
Reference 5 and 6 have been replaced with the newer studies. Please see Line 541-544 on Page 19.

Reviewer 3 Report (New Reviewer)
Comments and Suggestions for Authors
The article describes a monitoring system for swallowing disorders. The article is well-written and the topics is interesting. The authors have tested their system with clinical patients, which further proved the functionality of their monitoring system. One thing is missing is to discuss the used time length and stability of the sensor. Minor revision is suggested.
Author Response
Thank you very much for this important suggestion.Comment 1: The article describes a monitoring system for swallowing disorders. The article is well-written and the topics is interesting. The authors have tested their system with clinical patients, which further proved the functionality of their monitoring system. One thing is missing is to discuss the used time length and stability of the sensor. Minor revision is suggested.
Response 1:
The used time length and the stability of the sensors should be addressed more thoroughly through comprehensive clinical testing. We have listed this as important future work. Please see Lines 490-493 on page 18.
This manuscript is a resubmission of an earlier submission. The following is a list of the peer review reports and author responses from that submission.
Round 1
Reviewer 1 Report
Comments and Suggestions for Authors
In this paper, a monitoring and alert system for swallowing disorders is presented, utilizing noninvasive sensors to measure nasal airflow, surface electromyography signals, and thyroid cartilage movement. The system is tested on a group of healthy individuals and compared to a group of individuals with a history of unilateral stroke. The measurements showed significant differences between groups, indicating that the symptoms of swallowing disorders are detectable. They can be used for assessing the level of the disorder, suitable for performing screening, and for providing nutrient intake suggestions.
The manuscript is clearly written, and possible shortcomings compared to other monitoring methods are well addressed.
There is only one minor typo in line 17:
Please change “dis-eases” to “diseases.
Author Response
Reply for Reviewer 1:
Thank you very much for the recognition and comments. Here are the responses to each comment.
1. There is only one minor typo in line 17: Please change “dis-eases” to “diseases.
Reply: Thank you very much for your recognition and comments. The typo has been corrected. Please see Line 17 (p.1).

Reviewer 2 Report
Comments and Suggestions for Authors
In the manuscript, the authors have reported on a system capable of monitoring and alerting for swallowing disorders, utilizing three types of sensors. Overall, the quality of the work, content presentation, and scientific soundness are satisfactory. The manuscript can be accepted after addressing the following minor issues:
1. Include a brief statement of the selection criteria for participants involved in the test.
2. Correct the location of the ground electrode, whether on the chin or chest (the sentence in line 122 and Figure 2 do not match).
3. Provide more references to compare the patterns of the swallowing and respiration signals obtained in Figure 6 with more literature or commercial devices to validate the significance of the waveform obtained.
Comments on the Quality of English LanguageOverall, the quality of the English is satisfactory.
Author Response
Reply for Reviewer 2:
Thank you very much for the recognition and comments. Here are the responses to each comment.
1. Include a brief statement of the selection criteria for participants involved in the test.
Reply: Thanks for the reviewer’s comment. The participants of the test were selected from the rehabilitation department clinic at Chang Gung Memorial Hospital. Moreover, the inclusion criteria of the participants in two groups have been listed in Section 2.5. We add one more sentence at the beginning of Section 2.5 to explain where we selected the participants. Please see Lines 248-249 (p.8).
2. Correct the location of the ground electrode, whether on the chin or chest (the sentence in line 122 and Figure 2 do not match).
Reply: Thanks for the reviewer’s comment. The location of the ground electrode should be “on the neck near the chest” (Fig. 2b). The sentence in Line 121 has been corrected. Thank you very much.
3. Provide more references to compare the patterns of the swallowing and respiration signals obtained in Figure 6 with more literature or commercial devices to validate the significance of the waveform obtained.
Reply: Thanks for the reviewer’s comment. We add more references ([21]-[25]) and make Table 12 to compare the commercial devices (or sensors) with the sensors used in this paper. Please check Lines 456-461 (P.17) and Lines 573-584 (P.19).

Reviewer 3 Report
Comments and Suggestions for Authors
This manuscript introduces a promising swallow state monitoring system, incorporating a nasal airflow device, sEMG sensors, and a force-resistive sensor. The embedded board (Arduino uno and Ble module) efficiently collects and processes the data. The system is useful, but it is quite simple. There are some significant issues:
1. The system needs to be explained in detail. For example, how can the Arduino-embedded board read signals from sEMG and Nasal airflow devices?
2. The FSR, while functional, is not an ideal choice due to its lack of sensitivity and inherent stiffness. This necessitates a tight attachment to the neck, causing discomfort for the wearer. It's important to consider the user's comfort in the design choices. I don't think that FSR is the right sensor for this location.
3. The Bluetooth module seems to be an old version (classic Bluetooth), usually not used; the author should choose a more advanced embedded board (for example, Rasberry, Arduino 33 BLE, Adafruit, etc., with BLE Bluetooth).
4. To show the contribution of this work, a comparison table with other research/real commercial devices in the same field is needed.
5. A video of system operation is required.
In general, the unscientific selection of equipment leads to recording results that are not really outstanding and reliable. The manuscript also does not have a reasonable data processing/analysis method. Therefore, I assess that the manuscript does not have enough contributions to be published.
Comments on the Quality of English LanguageMinor editing of English language required
Author Response
Reply for Reviewer 3:
Thank you very much for the valuable comments. Here are the responses to each comment.
1. The system needs to be explained in detail. For example, how can the Arduino-embedded board read signals from sEMG and Nasal airflow devices?
Reply: Thank you very much for the comment. We add more descriptions to explain how the signal collector reads the signals from sEMG and nasal airflow devices. Please refer to Lines 169-174 (P.5).
2. The FSR, while functional, is not an ideal choice due to its lack of sensitivity and inherent stiffness. This necessitates a tight attachment to the neck, causing discomfort for the wearer. It's important to consider the user's comfort in the design choices. I don't think that FSR is the right sensor for this location.
Reply: Thank you very much for this comment. The sensor’s sensitivity and the user's comfort are the most important issues in this study. Thus, instead of attaching the FSR to the larynx surface, a throat-belt is used and the FSR is fixed on the center of the belt such that a participant can wear the belt around the neck (Figure 4). The belt is elastic with soft cushioned material and secured with Velcro straps. With a maximum width of 5 cm, this design ensures minimal discomfort for the wearer and reduces the need for the belt to be tightened excessively. Additionally, a small soft airbag is inserted between the belt and FSR, ensuring the FSR remains centered on the thyroid cartilage without movement during testing. When the belt is fastened around the neck, the airbag provides stable initial pressure on the FSR as the measurement baseline, ensuring sensor sensitivity.
We revised the descriptions about the usage of the FSR in Section 2.2.3 (before Figure 4). Please check Lines 140-158 (p.4).
3. The Bluetooth module seems to be an old version (classic Bluetooth), usually not used; the author should choose a more advanced embedded board (for example, Rasberry, Arduino 33 BLE, Adafruit, etc., with BLE Bluetooth).
Reply: Thank you very much for the comment. Regarding the comments about the Bluetooth module and the use of more advanced boards, we would like to clarify that our focus in this work was on creating a functional device rather than optimizing for low power consumption or utilizing advanced boards. Our primary goal was to ensure the device operated correctly. In future updates, we plan to enhance the device, making it more advanced and future-proof.
4. To show the contribution of this work, a comparison table with other research/real commercial devices in the same field is needed.
Reply: Thanks for the reviewer’s comment. We add more references ([21]-[25]) and make Table 12 to compare the commercial devices (or sensors) with the sensors used in this paper. Please check Lines 456-461 (P.17) and Lines 573-584 (P.19).
5. A video of system operation is required.
Reply: Thanks for the reviewer’s comment. We provide three videos to demonstrate
- Safe swallow without alerts: https://youtu.be/LFRf7facoks
- Unsafe swallow with the long swallowing time case (L_TET): https://youtu.be/7cgrQaVTdRU
- Unsafe swallow with the PMD case: https://youtu.be/cFhK9rixilY
